# Nonparametric Hyperbox Granular Computing Classification Algorithms

**Hongbing Liu [1,2,*], Xiaoyu Diao [2] and Huaping Guo [2]**

[1] Center of Computing, Xinyang Normal University, Xinyang 464000, China
[2] School of Computer and Information Technology, Xinyang Normal University, Xinyang 464000, China; xydiao2010@163.com (X.D.); hpguo_cm@163.com (H.G.)
[*] Correspondence: liuhbing@xynu.edu.cn

**Abstract:** Parametric granular computing classification algorithms lead to difficulties in terms of parameter selection, the multiple performance times of algorithms, and increased algorithm complexity in comparison with nonparametric algorithms. We present nonparametric hyperbox granular computing classification algorithms (NPHBGrCs). Firstly, the granule has a hyperbox form, with the beginning point and the endpoint induced by any two vectors in $N$-dimensional ($N$-D) space. Secondly, the novel distance between the atomic hyperbox and the hyperbox granule is defined to determine the joining process between the atomic hyperbox and the hyperbox. Thirdly, classification problems are used to verify the designed NPHBGrC. The feasibility and superiority of NPHBGrC are demonstrated by the benchmark datasets compared with parametric algorithms such as HBGrC.

**Keywords:** hyperbox granule; granular computing; distance; join operation

## 1. Introduction

The classification algorithm is a traditional data analysis method that is widely applied in many fields, including computer vision [1], DNA analysis [2], and physical chemistry [3]. For classification problems, the main method is the parameter-based learning method, whereby the relation between the input and the output is found to predict the class label of an input with unknown class label. The parameter-based learning method includes the analytic function method and the discrete inclusion relation method. The analytic function method establishes the mapping relationship between the input and output of the training datasets. The trained mapping is used to predict the class label of inputs with unknown class labels. Support Vector Machine (SVM) and multilayer perceptron (MLP) are kinds of methods by which linear or nonlinear mapping relationships are formed to predict the class label of inputs without class labels. The discrete inclusion relation method estimates the class labels of inputs based on the discrete inclusion relation between an input with a determined class label and an input without a class label and includes techniques such as random forest (RF) and granular computing (GrC). In this paper, we mainly study the classification algorithm using GrC, especially GrC with the form of hyperbox granule, the superiority and feasibility of which are shown in references [4–11].

As a classification and clustering method, GrC involves a computationally intelligent theory and method, and jumps back-and-forth between different granularity spaces [12–14]. Being fundamentally a data analysis method, GrC is commonly studied from the perspectives of theory and application, the latter of which includes pattern recognition, image processing, and industrial applications [12–19]. The main research issues of GrC include shape, operation, relation, granularity, etc.

A granule is a set of objects in which the elements are regarded to be objects with similar properties [17]. Binary granular computing proposes a conventional binary relation between two sets. Correspondingly, the operations between two sets are converted into the operation between

two granules, such as the intersection operation and the union operation between two sets (granules). Another research issue in GrC is how to define the distance between two granules. Chen and his colleague introduced Hamming distance to understand distance measurements with respect to binary granules for a rough set. Moreover, granule swarm distance is used to measure the uncertainty between two granules [20].

Operations between two granules are expressed as the equivalent form of membership grades, which are produced by the two triangular norms [15]. Kaburlasos defined the join operation and the meet operation as inducing granules with different granularity in terms of the theory of lattice computing [5,6]. Kaburlasos defined the fuzzy inclusion measure between two granules on the basis of the defined join operation and meet operation, and the fuzzy lattice reasoning classification algorithm was designed based on the distance between the beginning point and the endpoint of the hyperbox granule [7].

The relation between two granules is mainly used to generate the rules of association between inputs and outputs for classification problems and regression problems. A specialized version of this general framework is proposed by GrC theory in order to mine the potential relations behind data [21]. Kaburlasos and his colleague embed the lattice computing, including GrC, into a fuzzy inference system (FIS), and preliminary industrial applications have demonstrated the advantages of their proposed GrC methods [4].

Granularity is the index of measurement for the size of a granule and the means by which the granularity of a granule can be measured is one of the foundational issues in GrC. Yao regarded a granule as a set and defined the granularity as the cardinality of the set by a strictly monotonic function [14]. As a classification algorithm, GrC is concerned with human information processing procedures: the procedure includes both the data abstraction and the derivation of knowledge from information. To induce and deduce knowledge from the data, parameters are introduced to achieve suitable prior knowledge from the given data, such as the granularity threshold, the $\lambda$ of positive valuation function used for the construction of fuzzy inclusion measure between two granules, and the maximal number of data belonging the granule, thus resulting in some redundant granules during the training process. On one hand, these parameters improve the performance of GrC classification algorithms and GrC clustering algorithms. On the other hand, these parameters also have negative impacts, such as the higher time consumption required by parametric GrC compared with nonparametric GrC algorithms.

The proposed nonparametric hyperbox GrC has two main advantages for classification tasks. First, the nonparametric hyperbox GrC achieved better performance when compared with the parametric hyperbox GrC. Second, compared with the nonparametric hyperbox GrC, the parametric hyperbox GrC classification algorithms perform the algorithm multiple times, which is time-consuming for the selection of parameters, such as the parameter for positive valuation function and the threshold of granularity. The nonparametric hyperbox granular computing classification algorithm (NPHBGrC) includes the following steps. First, the granule has a regular hyperbox shape, with the beginning point and the endpoint that are induced by two vectors in *N*-dimensional *(N*-D) space; second, the distance between two hyperbox granules is introduced to determine their join process; and third, the NPHBGrC is designed and verified by the benchmark dataset compared with hyperbox granular computing classification algorithms (HBGrCs).

## 2. Nonparametric Granular Computing

In this section, we discuss the nonparametric granular computing, including the representation of granules, the operation between two granules, and the distance between two granules.

### 2.1. Representation of Hyperbox Granule

For granular computing in *N*-D space, we suppose a granule as a regular shape, such as a hyperbox with the beginning point $x$ and the endpoint $y$ which satisfy the partial order relation $x \preceq y$. The beginning point $x$ and the endpoint $y$ are vectors in *N*-D space, and the hyperbox granule has

the form $G = [Bp, Ep]$, where $Bp$ is the beginning point and $Ep$ is the endpoint. For any two vectors $x = (x_1, x_2, \ldots, x_N)$ and $y = (y_1, y_2, \ldots, y_N)$, if the two vectors $x$ and $y$ satisfy the partial order relation $x \preceq y$, then $Bp = x$ and $Ep = y$, otherwise $Bp = x \wedge y$ and $Ep = x \vee y$. The partial order relation between two vectors in *N*-D space is defined as follows

$$x \preceq y \Leftrightarrow x_1 \leq y_1, x_2 \leq y_2, \ldots, x_N \leq y_N$$

The operation $\wedge$ and operation $\vee$ between two vectors are defined as follows

$$x \wedge y = (x_1 \wedge y_1, x_2 \wedge y_2, \ldots, x_N \wedge y_N)$$
$$x \vee y = (x_1 \vee y_1, x_2 \vee y_2, \ldots, x_N \vee y_N)$$

where the operation $\wedge$ and operation $\vee$ between two scalars are $a \wedge b = \min\{a, b\}$ and $a \vee b = \max\{a, b\}$.

Obviously, for two vectors $x$ and $y$ in *N*-D space, we form the hyperbox granule with the form of vector $G = [Bp, Ep]$, where $Bp$ is the beginning point and $Ep$ is the endpoint of the granule. In the following sections, we represent hyperbox granule by $G = [x, y]$ for *N*-D space. In 2-D space, the granule $G = [x, y]$ is box, and in *N*-D space, the granule $G = [x, y]$ is a hyperbox.

### 2.2. Operations between Two Hyperbox Granules

For two hyperbox granules $G_1 = [x_1, y_1]$ and $G_2 = [x_2, y_2]$, the join hyperbox granule is the following form by the join operation

$$G_1 \vee G_2 = [x_1 \wedge x_2, y_1 \vee y_2] \tag{1}$$

where $x_1 = (x_{11}, x_{12}, \ldots, x_{1N})$ and $y_1 = (y_{11}, y_{12}, \ldots, y_{1N})$ are vectors, $x_1 \wedge x_2 = (x_{11} \wedge x_{21}, x_{12} \wedge x_{22}, \ldots, x_{1N} \wedge x_{2N})$, $y_1 \vee y_2 = (y_{11} \vee y_{21}, y_{12} \vee y_{22}, \ldots, y_{1N} \vee y_{2N})$.

The join hyperbox granule has greater granularity than the original hyperbox granules. The original hyperbox granules and the join hyperbox granule have the following relations.

$$G_1 \subseteq G_1 \vee G_2$$
$$G_2 \subseteq G_1 \vee G_2.$$

The meet hyperbox granule has the following form by the meet operation

$$G_1 \wedge G_2 = \begin{cases} [x_1 \vee x_2, y_1 \wedge y_2] & x_1 \vee x_2 \preceq y_1 \wedge y_2 \\ \varnothing & otherwise \end{cases}. \tag{2}$$

The meet hyperbox granule has less granularity than the original hyperbox granules. The meet hyperbox granule and the original hyperbox granules have the following relations.

$$G_1 \wedge G_2 \subseteq G_1$$
$$G_2 \subseteq G_1 \vee G_2.$$

For example, in 2-D space, $G_1 = [0.05, 0.15, 0.48, 0.68]$ and $G_2 = [0.1, 0.2, 0.5, 0.7]$ are two hyperbox granules, and their join hyperbox granule is $G_1 \vee G_2 = [0.05, 0.15, 0.5, 0.7]$, which is induced by the aforementioned join operation. These three hyperboxes are shown in Figure 1.

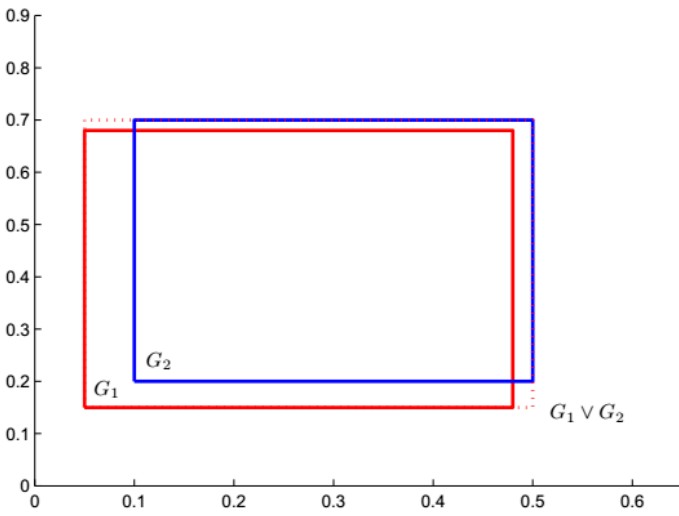

**Figure 1.** Join process of two hyperbox granules in 2-D space.

*2.3. Novel Distance between Two Hyperbox Granules*

The atomic hyperbox granule is a point in *N*-D space and is represented as the hyperbox with a beginning point and an endpoint which are identical. We can measure the distance relation between the point and the hyperbox granule.

For *N*-D space, the distance function is the mapping between *N*-D vector space and 1-D real space. From a visual point of view, distance is a numerical description of how far two objects are from one another. The distance function between two hyperbox granules is the mapping between hyperbox granule space and 1-D space, and a larger distance means that there is a smaller overlap area between the two hyperbox granules. The distance function between two hyperbox granules in granule space S is a function:

$$d : S \times S \to R$$

where *R* denotes the set of real numbers. We define the distance between two hyperbox granules $G_1 = [Bp_1, Ep_1]$ and $G_2 = [Bp_2, Ep_2]$ as follows.

**Definition 1.** *The distance between point P and hyperbox granule $G = [Bp, Ep]$ is defined as*

$$D(P, G) = d(P, Bp) + d(P, Ep) - d(Bp, Ep)$$

*where Bp is the beginning point and is denoted as $Bp = (x_1, x_2, \ldots, x_N)$, Ep is the endpoint and is denoted as $Ep = (y_1, y_2, \ldots, y_N)$, and $d(\cdot, \cdot)$ is the Manhattan distance between two points:*

$$d(Bp, Ep) = \|Bp - Ep\|_1 = |x_1 - y_1| + \ldots + |x_N - y_N|.$$

Suppose $P = (p_1, p_2, \ldots, p_N)$ is a point in *N*-D space, *G* is a hyperbox granule in granule space, the distance between *P* and *G* is the mapping between the granule space and the real space which satisfies the following non-negativity property.

$$
\begin{aligned}
D(P, G) &= d(P, Bp) + d(P, Ep) \\
&= |p_1 - x_1| + \ldots + |p_N - x_N| + |y_1 - p_1| + \ldots + |y_N - p_N| - (|y_1 - x_1| + \ldots + |y_N - x_N|). \\
&= (|p_1 - x_1| + |p_1 - y_1| - |y_1 - x_1|) + \ldots + (|p_N - x_N| + |p_N - y_N| - |y_N - x_N|) \geq 0
\end{aligned}
$$

The distance between the point and hyperbox granule *G* is explained in 2-D space. For $G = [0.1, 0.2, 0.4, 0.3]$ and the point $P(0.3, 0.4)$, $d(P, Bp) = 0.4$, $d(P, Ep) = 0.2$, $d(Bp, Ep) = 0.4$, $D(P, G) =$

$0.2 > 0$. The location of $P$ and $G$ is shown in Figure 2. As shown in Figure 2, the point $P$ is outside the hyperbox granule $G$.

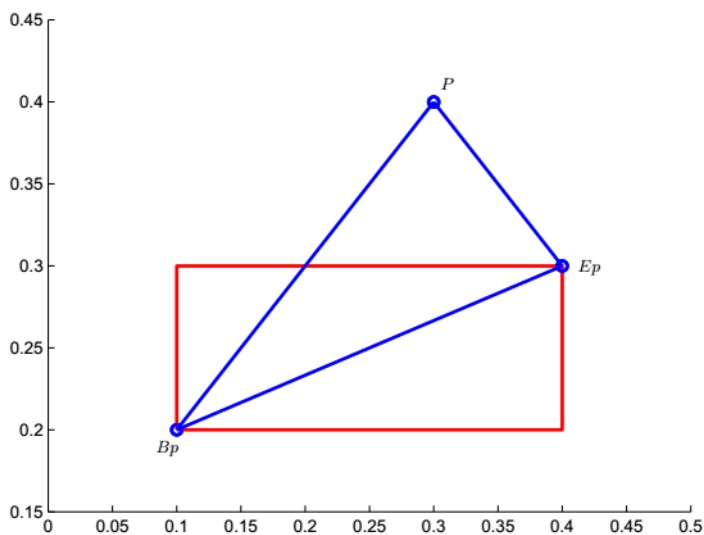

**Figure 2.** Distance between a point and a hyperbox granule in 2-D space.

**Theorem 1.** *In N-D space, the point $P$ is inside the hyperbox granule $G$ if and only if $D(P,G) = 0$.*

**Proof: Suppose** $Bp = (x_1, x_2, \ldots, x_N)$, $Ep = (y_1, y_2, \ldots, y_N)$, and $P = (p_1, p_2, \ldots, p_N)$.

If the point $P$ is inside the hyperbox granule $G = [Bp, Ep]$, then $Bp \preceq P$ and $P \preceq Ep$, $d(P, Bp) = p_1 - x_1 + p_2 - x_2 + \ldots + p_N - x_N$ and $d(P, Ep) = y_1 - p_1 + y_2 - p_2 + \ldots + y_N - p_N$:

$$
\begin{aligned}
&d(P, Bp) + d(p, Ep) \\
&= p_1 - x_1 + p_2 - x_2 + \ldots + p_N - x_N + y_1 - p_1 + y_2 - p_2 + \ldots + y_N - p_N \\
&= y_1 - x_1 + y_2 - x_2 + \ldots + y_N - x_N \\
&= d(Bp, Ep)
\end{aligned}
$$

Namely, $D(P, G) = d(P, Bp) + d(p, Ep) - d(Bp, Ep) = 0$

If $D(p, G) = 0$, then

$$
\begin{aligned}
D(P, G) &= d(P, Bp) + d(p, Ep) - d(Bp, Ep) \\
&= |p_1 - x_1| + |p_2 - x_2| + \ldots + |p_N - x_N| \\
&\quad + |y_1 - p_1| + |y_2 - p_2| + \ldots + |y_N - p_N| \\
&\quad - (|y_1 - x_1| + |y_2 - x_2| + \ldots + |y_N - x_N|) \\
&= (|y_1 - p_1| + |x_1 - p_1| - |y_1 - x_1|) \\
&\quad + (|y_2 - p_2| + |x_2 - p_2| - |y_2 - x_2|) \\
&\quad + \ldots \\
&\quad + (|y_N - p_N| + |x_N - p_N| - |y_N - x_N|) \\
&= 0
\end{aligned}
$$

Because $(|y_i - p_i| + |x_i - p_i| - |y_i - x_i|) \geq 0$ and $D(P, G) = 0$, $|y_i - p_i| + |x_i - p_i| - |y_i - x_i| = 0$. We discuss the relation between $x_i$ and $p_i$ and the relation between $y_i$ and $p_i$ in two situations.

When $p_i < x_i$, $p_i < y_i$ owing to $x_i \leq y_i$,

$$
|y_i - p_i| + |x_i - p_i| - |y_i - x_i| = y_i - p_i + x_i - p_i - y_i + x_i = 2(x_i - p_i) > 0
$$
$$
|y_i - p_i| + |x_i - p_i| - |y_i - x_i| = y_i - p_i + x_i - p_i - y_i + x_i = 2(x_i - p_i) > 0.
$$

Namely, $D(P, G) > 0$. This is obviously not in agreement with $D(P, G) = 0$, namely $x_i \leq p_i$. When $p_i > y_i$, $p_i > x_i$ owing to $x_i \leq y_i$,

$$|y_i - p_i| + |x_i - p_i| - |y_i - x_i| = p_i - y_i + p_i - x_i - y_i + x_i = 2(p_i - y_i) > 0.$$

Namely, $D(P, G) > 0$. This is obviously not in agreement with $D(P, G) = 0$, namely $p_i \leq y_i$. Therefore, $x_i \leq p_i$ and $p_i \leq y_i$, namely, $P$ is included in G. $\square$

**Definition 2.** *The distance between two hyperbox granules $G_1 = [Bp_1, Ep_1]$ and $G_2 = [Bp_2, Ep_2]$ is defined as*

$$D(G_1, G_2) = \max\{D(Bp_1, G_2), D(Ep_1, G_2)\}.$$

*Obviously, $D(G_1, G_2) \geq 0$ and the distance between two hyperbox granules have the following properties.*

**Theorem 2.** $D(G_1, G_2) = 0$ *if* $G_1 \subseteq G_2$.

**Proof:** Because $D(G_1, G_2) \geq 0$ and $D(G_1, G_2) = \max\{D(Bp_1, G_2), D(Ep_1, G_2)\} = 0$,
$D(Bp_1, G_2) = D(Ep_1, G_2) = 0$, according to Theorem 1, $Bp_1$ is inside the hyperbox granule $G_2$ and $Ep_1$ is inside the hyperbox granule $G_2$, namely $Bp_1 \in G_2$ and $Ep_1 \in G_2$. So $G_1 \subseteq G_2$.
If $G_1 \subseteq G_2$, both $Bp_1$ and $Ep_1$ are inside the hyperbox granule $G_2$. According to Theorem 1, $D(Bp_1, G_2) = 0$ and $D(Ep_1, G_2) = 0$, the maximum of $D(Bp_1, G_2)$ and $D(Ep_1, G_2) = 0$ is zero, namely,

$$D(G_1, G_2) = \max\{D(Bp_1, G_2), D(Ep_1, G_2)\} = 0.$$

$\square$

**Theorem 3.** $D(G_1, G_2) \neq D(G_2, G_1)$.

**Proof:**
$$D(G_1, G_2) = \max\{D(Bp_1, G_2), D(Ep_1, G_2)\}$$
$$= \max\{d(Bp_1, Bp_2) + d(Bp_1, Ep_2) - d(Bp_2, Ep_2),$$
$$d(Ep_1, Bp_2) + d(Ep_1, Ep_2) - d(Bp_2, Ep_2)\}$$

$$D(G_2, G_1) = \max\{D(Bp_2, G_1), D(Ep_2, G_1)\}$$
$$= \max\{d(Bp_2, Bp_1) + d(Bp_2, Ep_1) - d(Bp_1, Ep_1),$$
$$d(Ep_2, Bp_1) + d(Ep_2, Ep_1) - d(Bp_1, Ep_1)\}$$

Owing to $d(Bp_1, Ep_1) \neq d(Bp_2, Ep_2)$, $D(G_1, G_2) \neq D(G_2, G_1)$. $\square$

### 2.4. Nonparametric Granular Computing Classification Algorithms

For classification problem, the training set is the set $S$ and the NPHBGrC are proposed by the following steps to form the granule set $GS$, which is composed of hyperbox granules. First, the sample is selected to form the atomic hyperbox granule randomly. Second, the other sample with the same class label as the hyperbox granule in $GS$ is selected to form the join hyperbox by join operation. Third, the hyperbox granule is updated if the join hyperbox granule does not include the sample with the other class label. The NPHBGrC algorithms include training process and testing process, which are listed as Algorithms 1 and 2.

---

**Algorithm 1: Training process**

---

Input: Training set $S$
Output: Hyperbox granule set $GS$, the class label $lab$ corresponding to $GS$
    S1. Initialize the hyperbox granule set $GS = \varnothing$, $lab = \varnothing$;
    S2. $i = 1$;
    S3. Select the samples with class labels $i$, and generate set $X$;
    S4. Initialize the hyperbox granule set $GSt = \varnothing$;
    S5. If $GSt = \varnothing$, the sample $x_j$ in $X$ is selected to construct the corresponding atomic hyperbox granule $G_j$, $x_j$
is removed from $X$, otherwise $j = 1$;
    S6. The sample $x_k$ is selected from $X$ and forms the hyperbox granule $G_k$;
    S7. If the join hyperbox granule $G_j \vee G_k$ between $G_j$ and $G_k$ does not include the other class sample, the $G_j$ is
replaced by the join hyperbox granule $G_j \vee G_k$ and the samples included in $G_j \vee G_k$ with the class labels $i$ are
removed from $X$, namely, $G_j = G_j \vee G_k$, otherwise $GS$ and $lab$ are updated, $GS = GS \cup \{G_k\}$, $lab = lab \cup \{i\}$;
    S8. $j = j + 1$;
    S9. If $i = n$, output $GS$ and class label $lab$, otherwise $i = i + 1$.

---

**Algorithm 2: Testing process**

---

Input: inputs of unknown datum $x$, the trained hyperbox granule set $GS$ and class label $lab$
Output: class label of $x$
    S1. For $i = 1 : |GS|$;
    S2. Compute the distance $D(x, G_i)$ between $x$ and $G$ in $GS$;
    S3. Find the minimal distance $D(x, G_i)$;
    S4. Find the corresponding class label of the $G_i$ as the label of $x$.

---

We take the training set including 10 training data for example to explain the training algorithm. Suppose the training set is

$$S = \{(x_1, y_1), (x_2, y_2), (x_3, y_3), (x_4, y_4), (x_5, y_5), (x_6, y_6), (x_7, y_7), (x_8, y_8), (x_9, y_9), (x_{10}, y_{10})\}.$$

where the inputs of data are

$$x_1 = (4, 7), x_2 = (7, 6), x_3 = (8, 2), x_4 = (2, 4), x_5 = (5, 5),$$
$$x_6 = (5, 9), x_7 = (6, 4), x_8 = (5, 7), x_9 = (7, 3), x_{10} = (3, 7)$$

The corresponding class label is

$$y_1 = 1, y_2 = 1, y_3 = 2, y_4 = 2, y_5 = 2, y_6 = 1, y_7 = 2, y_8 = 1, y_9 = 2, y_{10} = 2.$$

We explain the generation of $GS$ by Algorithm 1. The $x_1 = (4, 7)$ is selected to form the atomic hyperbox granule $G_1 = [4, 7, 4, 7]$ with the granularity 0 and the class label 1 shown in Figure 3a. The second datum $x_2 = (7, 6)$ with the same class label as $G_1$ is selected to generate the atomic hyperbox granule $[7 \, 6 \, 7 \, 6]$ which is joined with $G_1$ and forms the join hyperbox granule $[x_2, x_2] \vee G_1 = [4, 6, 7, 7]$. Since there are no data with the other class label lying in the join hyperbox granule $[x_2, x_2] \vee G_1 = [4, 6, 7, 7]$, the $G_1$ is replaced by the join hyperbox granule, namely $G_1 = [x_2, x_2] \vee G_1 = [4, 6, 7, 7]$, as shown in Figure 3b. The third datum x6 with the same class label with $G_1$ is selected to generate atomic hyperbox granule $[x_6, x_6] = [5, 9, 5, 9]$, which is joined with $G_1$ and forms the join hyperbox granule $[x_6, x_6] \vee G_1 = [4, 6, 7, 9]$. As there are no data with the other class label lying in the join hyperbox granule $[x_6, x_6] \vee G_1 = [4, 6, 7, 9]$, $G_1$ is replaced by $[x_6, x_6] \vee G_1 = [4, 6, 7, 9]$, namely $G_1 = [4, 6, 7, 9]$, as shown in Figure 3c. During the join process, the datum with the class label with the hyperbox granule lies in the hyperbox granule is not considered the join process, such as datum x8 with the class label 1. In this way, the hyperbox granule $G_1 = [4, 6, 7, 9]$ with the blue lines is generated for the data with the class label 1. The same strategy is adopted for the data with the class label 2;

two hyperbox granules $G_2 = [2, 2, 8, 5]$ and $G_3 = [3, 7, 3, 7]$ are generated and are shown in Figure 3d. For the training set $S$, the achieved granule set is $GS = \{G_1, G_2, G_3\}$ and the corresponding class label is $lab = \{1, 2, 2\}$. The granules in GS are shown in Figure 3d; the granule marked with the blue lines is the granule with class label 1, and the granules with the red lines are the granules with class label 2.

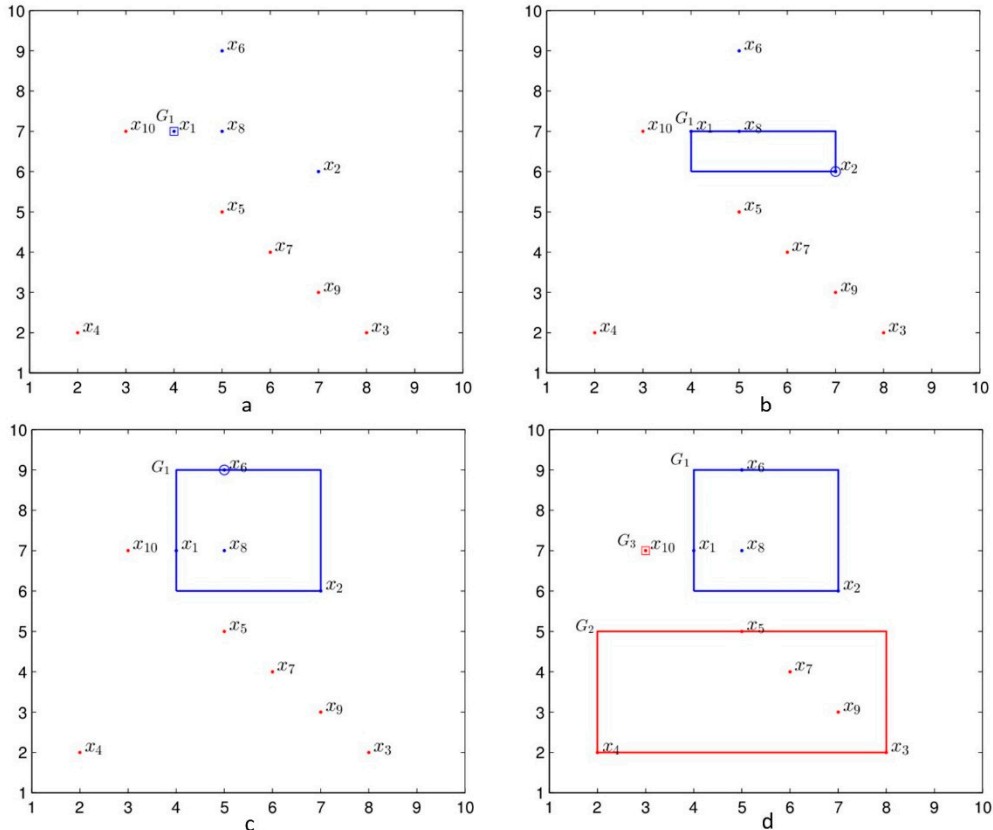

**Figure 3.** The example of Algorithm 1. (**a**) The hyperbox granule by $x_1$., (**b**) the join hyperbox granule between the granule $G_1$ and the atomic hyperbox granule [7 6 7 6], (**c**) the join hyperbox granule between $G_1$ and the atomic hyperbox granule [5 9 5 9], and (**d**) the granule set including three hyperbox granules with class label 1 (blue) and class label 2 (red).

## 3. Experiments

The effectiveness of the NPHBGrC is evaluated with a series of empirical studies including the classification problems in 2-D space and classification problems in *N*-D space. We compare NPHBGrC with GrC with parameters, such as the HBGrC [22], and evaluate the performance of classification algorithms by the threshold of granularity of HBGrC(Par.), the number of hyperbox granules (Ng), time cost (T(s)) including the training and testing processes, training accuracy (TAC), and testing accuracy (AC).

### 3.1. Classification Problems in 2-D Space

In the first benchmark study, the two spiral curve classification problem [23], Ripley classification problem [24], and sensor2 classification problem (wall—following robot navigation data) from the websites http://archive.ics.uci.edu/ml/datasets.html, which were created in two dimensions, were used to assess the efficacy of classification algorithms and to visualize the boundary of classification. The details of the datasets and classification performance are summarized in Table 1. The number of training data (#Tr), the number of testing data (#Ts), and the performances of NPHBGrC and HBGrC are shown in Table 1. From Table 1, it can be seen that NPHBGrC has greater or equal testing accuracies and less time cost compared with HBGrC. NPHBGrC has less time cost than HBGrC due to the fact

that HBGrC produces some redundant hyperbox granules. Figures 4 and 5 show the boundaries of NPHBGrC and HBGrC for the Ripley dataset.

**Table 1.** The classification problems and their performances in 2-D space.

| Dataset | #Tr | #Ts | Algorithms | Par. | Ng | TAC | AC | T(s) |
|---------|-----|-----|------------|------|-----|-----|-------|--------|
| Spiral | 970 | 194 | NPHBGrC | – | 58 | 100 | 99.48 | 0.6864 |
|         |     |     | HBGrC | 0.08 | 161 | 100 | 99.48 | 1.6380 |
| Ripley | 250 | 1000 | NPHBGrC | – | 32 | 100 | 90.2 | 0.0625 |
|        |     |      | HBGrC | 0.27 | 67 | 96 | 90.1 | 0.1159 |
| Sensor2 | 4487 | 569 | NPHBGrC | – | 4 | 100 | 99.47 | 1.0764 |
|         |      |     | HBGrC | 4 | 8 | 100 | 99.47 | 1.365 |

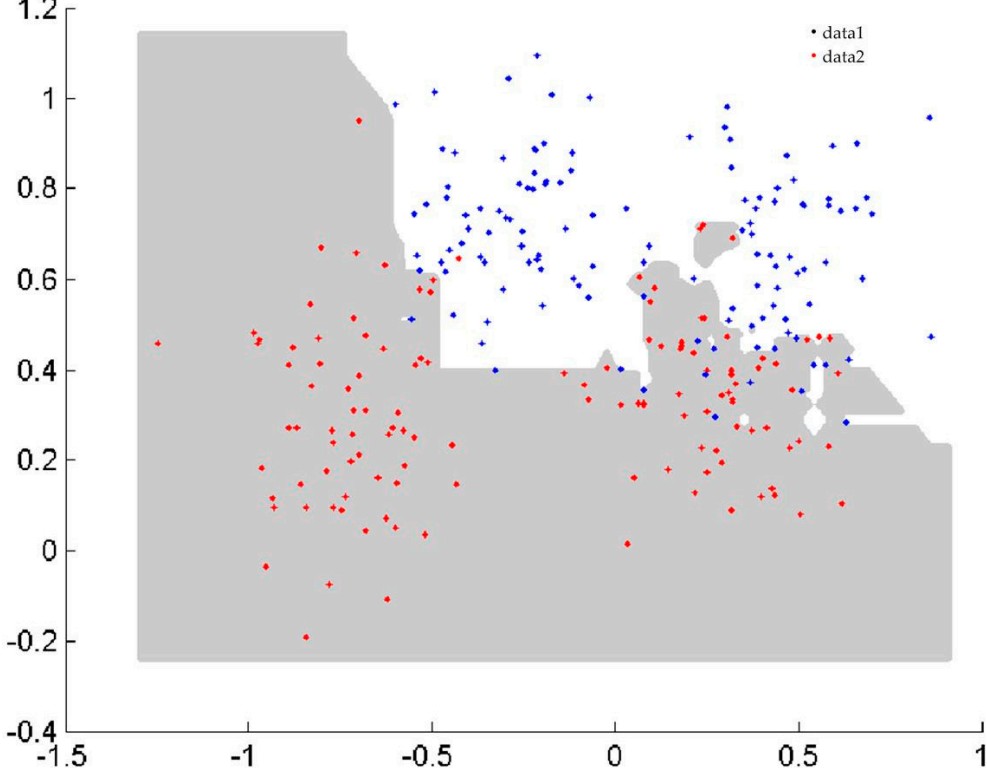

**Figure 4.** Boundary performed by nonparametric hyperbox granular computing classification algorithm (NPHBGrC) for the Ripley dataset.

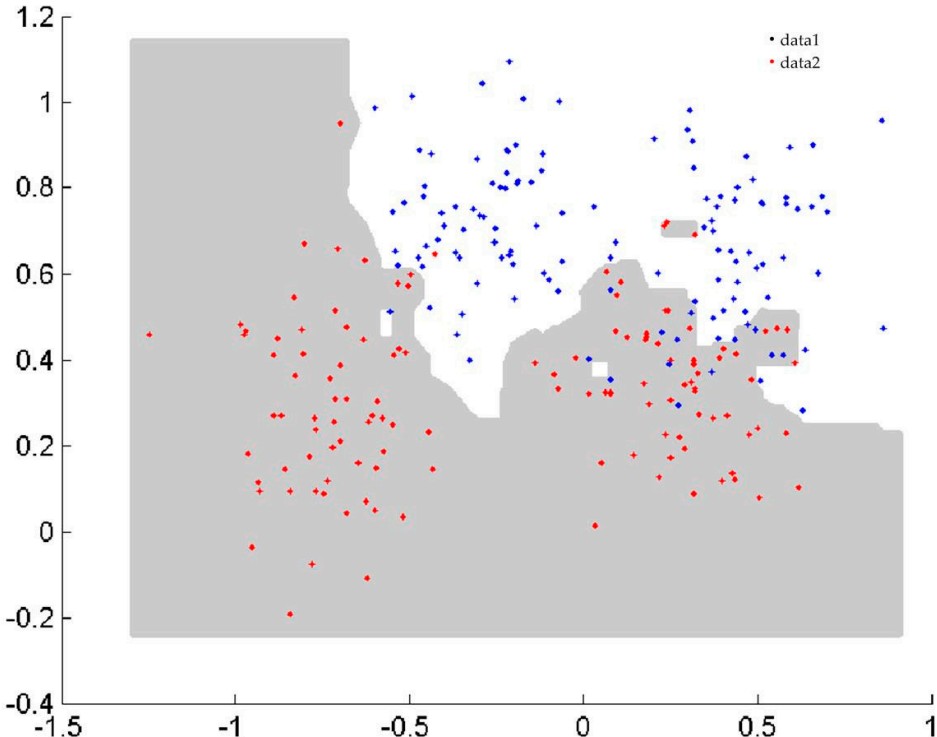

**Figure 5.** Boundary performed by hyperbox granular computing classification algorithm (HBGrC) for the Ripley dataset.

### 3.2. Classification Problems in N-dimensional (N-D) Space

In this section, we verify the performance of the proposed classification algorithms which are extended to *N*-D space compared with the HBGrC by the selected benchmark datasets from the website, http://archive.ics.uci.edu/ml/. These datasets are the most popular datasets since 2007, and the characteristics and the performance of the datasets are listed in Tables 2 and 3.

**Table 2.** The classification problems in *N-dimensional (N*-D) space.

| Datasets | N | Classes | Samples |
|----------|------|---------|---------|
| *Iris* | 4 | 3 | 150 |
| *Wine* | 13 | 3 | 178 |
| *Phoneme* | 5 | 2 | 5404 |
| *Sensor4* | 4 | 4 | 5456 |
| *Car* | 6 | 5 | 1728 |
| *Cancer2* | 30 | 2 | 532 |
| *Semeion* | 256 | 10 | 1593 |

For the parametric algorithm, in order to facilitate the selection of parameters of thresholds of granularities, the $R^N$ space is normalized into the $[0,1]^N$ space, the granularity parameters are set to between 0 and 0.5 with steps of 0.01 for the n-class classification problems performed by HBGrC.

A 10-fold cross-validation is used to evaluate the parametric and nonparametric classification algorithms. For each dataset, the nonparametric and parametric algorithms are performed for each fold, and the parametric algorithms are performed 51 times for each fold due to the selection of granularity threshold parameters.

The performances of classification algorithms include the maximal testing accuracies, the mean testing accuracies, the minimal testing accuracies, and the standard deviation of testing accuracies.

The superiority of algorithms is evaluated by the mean testing accuracies and the stability of algorithms is verified by the standard deviation of testing accuracies, which are shown in Table 3. From the Table 3, it can be seen that NPHBGrC algorithms are superior to HBGrC algorithms regardless of the maximum testing accuracy (max), the mean testing accuracy (mean), or the minimum testing accuracy (min). On the other hand, it can also be seen from Table 3 that the standard deviations of 10-fold cross-validation by NPHBGrC are less than those of HBGrC, which shows that NPHBGrC algorithms are more stable than HBGrC algorithms.

The testing accuracies are the main evaluation indices for the classification algorithms. A *t*-test was used to verify the testing accuracies by nonparametric algorithms and parametric algorithms statistically. If $h = 0$, then the testing accuracies achieved by NPHBGrC and HBGrC have no significant difference statistically, although $h = 0$, but $p$ is relatively small, close to 0.05, we regard the achieved testing accuracies have significant difference. If $h = 1$, then the testing accuracies achieved by NPHBGrC and HBGrC are significantly different, and we can illustrate the superiority of the algorithm by the mean testing accuracy, especially, although $h = 1$, but $p$ is relatively small, close to 0.05, we regard the achieved testing accuracies as having no significant difference.

For the datasets *Iris*, *Wine*, *Cancer1*, *Sensor4*, and *Cancer2*, $h = 0$, as shown in Table 4. Statistically, the testing accuracies obtained by NPHBGrC and HBGrC have no significant difference from the $h$ values of *t*-test listed in Table 3, and the testing accuracies of NPHBGrC are slightly higher than those of HBGrC in terms of maximal testing accuracies, mean testing accuracies, and the minimal testing accuracies listed in Table 3.

For the datasets Phoneme, Car, and Semeion, $h = 1$, as shown in Table 4. Statistically, the testing accuracies by NPHBGrC and HBGrC are significantly different, and we determine which is the better classification algorithm for NPHBGrC and HBGrC from the mean testing accuracies in Table 3. NPHBGrC algorithms are better than HBGrC algorithms, since the mean testing accuracies obtained by NPHBGrC are greater than those obtained by HBGrC, as shown in Table 3.

The computational complexities are evaluated by the time cost, including the training and testing time cost. Obviously, NPHBGrC algorithms have lower computational complexities compared with HBGrC due to the redundant hyperbox granules and the parameter selection for HBGrC.

**Table 3.** The performances in *N*-D space.

| Dataset | Algorithms | Testing Accuracy | | | | T(s) |
| --- | --- | --- | --- | --- | --- | --- |
| | | max | mean | min | std | |
| *Iris* | NPHBGrC | 100 | 98.6667 | 93.3333 | 3.4427 | 0.0265 |
| | HBGrC | 100 | 97.3333 | 93.3333 | 2.8109 | 1.1560 |
| *Wine* | NPHBGrC | 100 | 96.8750 | 93.7500 | 3.2940 | 0.0406 |
| | HBGrC | 100 | 96.2500 | 87.5000 | 4.3700 | 1.0140 |
| *Phoneme* | NPHBGrC | 91.6512 | 89.8236 | 88.3117 | 1.1098 | 22.4844 |
| | HBGrC | 87.5696 | 85.9350 | 83.1169 | 1.3704 | 422.3009 |
| *Cancer1* | NPHBGrC | 100 | 98.5075 | 95.5224 | 1.7234 | 0.9064 |
| | HBGrC | 100 | 97.6362 | 92.5373 | 2.6615 | 69.8214 |
| *Sensor4* | NPHBGrC | 100 | 99.4551 | 97.4217 | 0.8621 | 1.0670 |
| | HBGrC | 100 | 99.2157 | 96.6851 | 0.9944 | 71.8509 |
| *Car* | NPHBGrC | 97.6608 | 91.1445 | 81.8713 | 5.3834 | 8.7532 |
| | HBGrC | 94.7368 | 85.9593 | 77.7778 | 5.5027 | 1166.5 |
| *Cancer2* | NPHBGrC | 100 | 98.0769 | 92.3077 | 2.3985 | 0.4602 |
| | HBGrC | 100 | 97.4159 | 94.2308 | 1.9107 | 7.5676 |
| *Semeion* | NPHBGrC | 100 | 98.7512 | 97.4026 | 0.7177 | 6.7127 |
| | HBGrC | 97.4026 | 94.9881 | 92.2078 | 1.4397 | 533.2691 |

**Table 4.** The *t*-test values of comparison of NPHBGrC and HBGrC.

| Algorithms | *Iris* | | *Wine* | | *Phoneme* | | *Cancer1* | |
|---|---|---|---|---|---|---|---|---|
| | *h*-value | *p*-value | *h*-value | *p*-value | *h*-value | *p*-value | *h*-value | *p*-value |
| NPHBGrC-HBGrC | 0 | 0.3553 | 0 | 0.7222 | 1 | 0 | 0 | 0.3963 |
| **Algorithms** | *Sensor4* | | *Car* | | *Cancer2* | | *Semeion* | |
| | *h*-value | *p*-value | *h*-value | *p*-value | *h*-value | *p*-value | *h*-value | *p*-value |
| NPHBGrC-HBGrC | 0 | 0.5722 | 1 | 0.0472 | 0 | 0.5041 | 1 | 0 |

### 3.3. Classification for Imbalanced Datasets

For the imbalanced datasets, an imbalanced dataset called yeast, including 1484 data, was used to verify the performance of the proposed algorithm, where the positive data belong to class NUC (class label 1 in the paper) and the negative data belong to the rest (class label 2 in the paper). The dataset can be downloaded from the website http://keel.es/. Five-fold cross-validation was used to evaluate the performance of NPHBGrC and HBGrC, such as the testing accuracy and class-based testing accuracy. The accuracies are listed in Table 5, and the histogram of accuracies is shown in Figure 6. For the testing set, AC is the total accuracy, C1AC is the accuracy of data with class label 1, and C2AC is the accuracy of data with class label 2. For the five tests, named Test 1, Test 2, Test 3, Test 4, and Test 5, NPHBGrC achieved better total accuracies (AC) than HBGrC for the imbalanced class problem yeast. The geometric mean (GM) of the true rates is defined in [22] and attempts to maximize the accuracy of each of the two classes with a good balance. From Table 5, it can be seen that the GM of NPHBGrC is 74.2023, which is superior to the GM of HBGrC (64.8344), and to the fuzzy rule-based classification systems (69.66) by Fernández [25] and the weighted extreme learning machine (73.19) by Akbulut [26].

**Table 5.** Performance of NPHBGrC and HBGrC for the imbalanced dataset "yeast".

| Tests | AC (%) | | C1AC (%) | | C2AC (%) | | G (%) | |
|---|---|---|---|---|---|---|---|---|
| | NPHBGrC | HBGrC | NPHBGrC | HBGrC | NPHBGrC | HBGrC | NPHBGrC | HBGrC |
| Test 1 | 78.7879 | 76.4310 | 58.1395 | 54.6512 | 87.2038 | 85.3081 | 75.4026 | 68.2802 |
| Test 2 | 74.0741 | 73.7374 | 48.8372 | 44.1860 | 84.3602 | 85.7820 | 72.8299 | 61.5659 |
| Test 3 | 76.7677 | 74.0741 | 55.8140 | 54.6512 | 85.3081 | 81.9905 | 74.7530 | 66.9394 |
| Test 4 | 74.0741 | 73.4007 | 51.1628 | 47.6744 | 83.4123 | 83.8863 | 74.7382 | 63.2395 |
| Test 5 | 76.0135 | 74.6622 | 57.6471 | 48.2353 | 83.4123 | 85.3081 | 73.2875 | 64.1472 |
| mean | 75.9435 | 74.4611 | 54.3201 | 49.8796 | 84.7393 | 84.4550 | 74.2023 | 64.8344 |

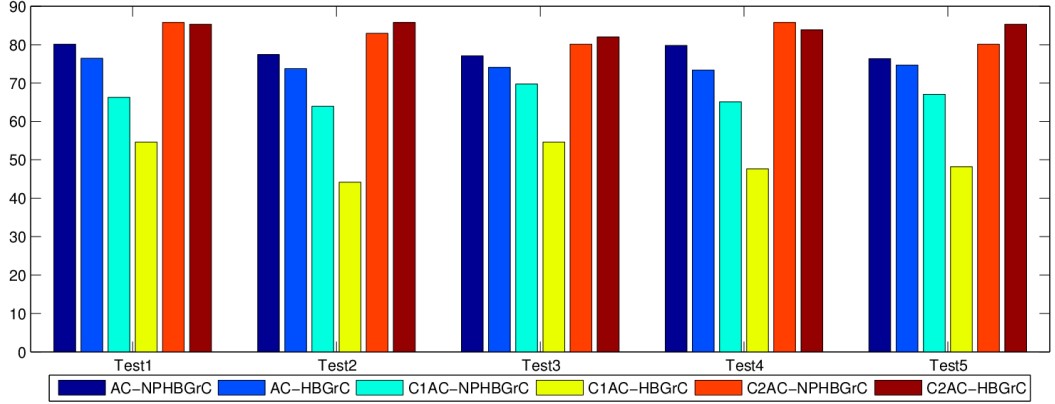

**Figure 6.** The histogram of performance by nonparametric hyperbox granular computing classification algorithm (NPHBGrC) and HBGrC for the yeast dataset.

## 4. Conclusions

According to the computational complexity produced the redundant hyperbox granules, we presented the NPHBGrC. The novel distance was introduced to measure the distance between two hyperbox granules and to determine the join process between two hyperbox granules. The feasibility and superiority of NPHBGrC were demonstrated by the benchmark datasets compared with HBGrC. There are some improvements in the NPHBGrC, for example, relating to the overfitting problem and the effect of the data order on the classification accuracy. The purpose of using distance in this paper was to determine the positional relationship between points (such as the points inside and outside the hyperbox) and the hyperbox. For the interval set and the fuzzy set, the operations between two granules were designed based on the fuzzy relation between two granules. For the fuzzy set, further research is needed in the future to determine how to use the proposed distance between two granules to design classification algorithms. For the classification of imbalanced datasets, the superiority of NPHBGrC was verified by the yeast dataset. In the future, the superiority and feasibility of GrC need to be verified using more metrics—such as the receiver operating curve (ROC), usually known as area under curve (AUC)—by more imbalanced datasets, and the computing theory of GrC needs further study for imbalanced datasets to achieve a better performance.

**Author Contributions:** Conceptualization, H.L.; Methodology, H.L. and X.D.; Validation, H.L., X.D., and H.G.; Data Curation, X.D.; Writing—Original Draft Preparation, H.L.

**Funding:** This work was supported in part by the Henan Natural Science Foundation Project (182300410145, 182102210132).

**Conflicts of Interest:** The authors declare no conflict of interest.

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
