# Peer review of "Nonparametric Hyperbox Granular Computing Classification Algorithms"

_information, doi:10.3390/info10020076_

Round 1

Reviewer 1 Report

- Although written english is understandable, it needs to be reviewed by a native speaker or professional.

- Authors must fix this part fo the text: " meet operator to induce the granules with different granularity Combined the theory of lattice computing ".

- Quality of figures should be improved.

- Authors should at least compare results with another hyperbox clustering technique.

- In table 2, the header "outputs" should be "classes".

- Authors should also add a table detailing class-based accuracy, not only the total for all classes in each dataset as this could hide problems with class imbalances.

- Results tables should be better presented as they are hard to read in their present state, a better format would fix this.

- A more profound discussion of experimentations results it missing.

- The conclusion is too short and not very insightful.

Author Response

Point 1: Although written english is understandable, it needs to be reviewed by a native speaker or professional.

Response 1: Thank you very much. The manuscript has been edited for proper English language, grammar, punctuation, spelling, and overall style by the highly qualified native English speaking editors at American Journal Experts.

Point 2:  Authors must fix this part for the text: " meet operator to induce the granules with different granularity Combined the theory of lattice computing ".

Response 2: Thank you very much, this part is replaced by “V. G. Kaburlasos defined the join operation and the meet operation to induce the granules with different granularity in term of  the theory of lattice computing [14]”

Point 3:  Quality of figures should be improved.

Response 3: Thank you very much. We upload the figures with eps formats.

Point 4:  Authors should at least compare results with another hyperbox clustering technique.

Response 4: Thank you very much. The comparisons are shown in Table 1, Table3, Figure 3 and Figure 4, The t-test results are shown in table 5.

Point 5:  In table 2, the header "outputs" should be "classes".

Response 5: Thank you very much, we revised it.

Point 6:  Authors should also add a table detailing class-based accuracy, not only the total for all classes in each dataset as this could hide problems with class imbalances.

Response 6: Thank you very much.  We added the sub-section “Classification for the imbalanced data sets”. For the classification of imbalanced data sets, the superiority of NPHBGrC is verified by the data set yeast, the superiority and feasibility of GrC need to be verified from the more metrics, such as the receiver operating curve (ROC) usually known as area under curve (AUC), by more imbalanced data sets in the future, and the computing theory of GrC achieving better performance needs further study for the imbalanced data sets.

Point 7:  Results tables should be better presented as they are hard to read in their present state, a better format would fix this.

Response 7: Thank you very much, we revised them.

Point 8: A more profound discussion of experimentations results it missing.

Response 8: Thank you very much, we revised them.

Point 9: The conclusion is too short and not very insightful.

Response 9: Thank you very much, we revised them.

Reviewer 2 Report

The manuscript proposes a granular computing based classification algorithm. The paper combines the granular computing and classification, which seems interesting, although in its present form I find it unacceptable for publication.

Below I give a non-exhaustive list of my concerns.

1. No motivation why the proposed algorithm is needed. Classification is a very classical problem. There are many existing algorithms to solve classification problems. What type of questions or scenarios motivate the proposed classification algorithm?

2. The proposed algorithm is not described clearly in section 2.4. A simple and small example may be helpful to explain the proposed algorithm. The focus of the paper is this classification algorithm, but the authors only use less than one page on this algorithm. The process of the algorithm should be explained in detail. Just listing pseudocode is not enough.

3. As for the experiments, the authors should compare the proposed algorithm with classical classification algorithms, such as decision trees, k-nearest neighbors, etc. Only comparing with parametric granular computing classification algorithm is not enough to show that the proposed algorithm is efficient or better.  

4. There is a serious problem with the linguistic quality of the manuscript. My impression is that is not only due to the authors' language skills, but also because the paper has been put together hastily and sloppily, as evidenced by the following errors 

line 9 lead -> leads

line 25 52, 57 granular computing or Granular Computing, should be consistent

line 43 granularity Combined 

line 69 does the parametric case

line 102 alignment

section 2.3 the point P italic or not in the equations is not consistent

line 185 191 alignment

the table formats 

Author Response

Response to Reviewer 2 Comments

Point 1: No motivation why the proposed algorithm is needed. Classification is a very classical problem. There are many existing algorithms to solve classification problems. What type of questions or scenarios motivate the proposed classification algorithm?

Response 1: Thank you very much, we summary the classification method as follows.

For the classification problems, the main method is the parameter-based learning method by which the relation between input and output is found to predict the class label of the input with unknown class label.  The parameter-based learning method includes the analytic function method and the discrete inclusion relation method. The analytic function method is to establish the mapping relationship between input and output for the training data sets. The trained mapping is used to predict the class label of the input with unknown class label. Support Vector Machine (SVM) and MultiLayer Perceptron (MLP) are the kind of method by which the linear or nonlinear mapping relationships are formed to predict the class label of the input without class label. The discrete inclusion relation method estimates the class label of input by the discrete inclusion relation between input with determined class label and input without class label, such as Random forest (RF) and granular computing (GrC). In the paper, we mainly study the classification algorithm by GrC, especially the GrC with the form of hyperbox granule whose superiority and feasibility are shown in the reference [4-11].

Point 2: The proposed algorithm is not described clearly in section 2.4. A simple and small example may be helpful to explain the proposed algorithm. The focus of the paper is this classification algorithm, but the authors only use less than one page on this algorithm. The process of the algorithm should be explained in detail. Just listing pseudocode is not enough.

Response 2: Thank you very much.

We take the training set including 10 training data for example to explain the training algorithm. Suppose the training set is

Where the inputs of data are

The corresponding class label is

We explain the generation of  by algorithm 1. The  is selected to form the atomic hyperbox granule  with the granularity 0 and the class label 1 shown in Fig.3 (a). The second datum  with the same class label as  is selected to generate the atomic hyperbox granule [7 6 7 6] which is joined with  and form the join hyperbox granule . Because there are not data with the other class label lying in the join hyperbox granule , the  is replaced by the join hyperbox granule, namely  shown in Fig.3(b). The third datum x6 with the same class label with  is selected to generate atomic hyperbox granule which is joined with  and form the join hyperbox granule . Because there are not data with the other class label lying in the join hyperbox granule,  is replaced by the , namely  shown in Fig.3(c). During the join process, the datum with the class label with the hyperbox granule lies in the hyperbox granule is not considered the join process, such as the datum x8 with the class label 1. In this way, the hyperbox granule  with the blue lines is generated for the data with the class label 1.  The same strategy is adopted to the data with the class label 2, two hyperbox granules  and  are generated and shown in Fig.3(d). For the training set S, the achieved granule set is  and the corresponding class label is , the granules in GS are shown in Fig.3(d), the granule marked with the blue lines is the granule with class label 1, the granules with the red lines are the granules with class label 2.

Point 3: As for the experiments, the authors should compare the proposed algorithm with classical classification algorithms, such as decision trees, k-nearest neighbors, etc. Only comparing with parametric granular computing classification algorithm is not enough to show that the proposed algorithm is efficient or better.  

Response 3: Thank you very much. Granular computing has been compared with classical classification algorithms, such as decision trees, k-nearest neighbors, and the superiority and feasibility of granular computing have been demonstrated, it is reasonable to compare the proposed granular computing with granular computing.

Point 4: There is a serious problem with the linguistic quality of the manuscript. My impression is that is not only due to the authors' language skills, but also because the paper has been put together hastily and sloppily, as evidenced by the following errors 

line 9 lead -> leads

line 25 52, 57 granular computing or Granular Computing, should be consistent

line 43 granularity Combined 

line 69 does the parametric case

line 102 alignment

section 2.3 the point P italic or not in the equations is not consistent

line 185 191 alignment

the table formats 

Response 4: thank you very much, we revised them.

Reviewer 3 Report

The paper presents an interesting work describing a nonparametric hyperbox granular computing classification algorithm. To publish this paper on this journal, I think the following concerns must be addressed.

1.In Section 2.3, the authors mentioned a novel distance calculation method between two hyperbox granules, which is a fundamental part of this paper. The authors only reviewed two distance calculation methods for rough set granules. How about interval set and fuzzy set granules? The authors should look into more related literature and explain the reason behind proposing the novel distance calculation method.

2In Section 3, the tables are unreadable and confused. Especially, why there are four values of accuracy for one data set? Besides, more explanations are expected for the experimental results.

3. Please check for consistency of formulas: italics, boldface, etc.

Author Response

Response to Reviewer 3 Comments

Point 1: In Section 2.3, the authors mentioned a novel distance calculation method between two hyperbox granules, which is a fundamental part of this paper. The authors only reviewed two distance calculation methods for rough set granules. How about interval set and fuzzy set granules? The authors should look into more related literature and explain the reason behind proposing the novel distance calculation method.

 Response 1: Thank you very much. The comment will inspire us to further explore the nature of the distance between two granules. The purpose of distance in this paper is to determine the position relationship between point and hyperbox, such as the points inside and outside the hyperbox. For the interval set and the fuzzy set, the operations between two granules are designed based on the fuzzy relation between two granules. For the fuzzy set, how to use the proposed distance between two granules to design the classification algorithms needs further research in the future.

Point 2: In Section 3, the tables are unreadable and confused. Especially, why there are four values of accuracy for one data set? Besides, more explanations are expected for the experimental results.

Response 2: Thank you very much.

The performances of classification algorithms include the maximal testing accuracies, the mean testing accuracies, the minimal testing accuracies, and the standard deviation of testing accuracies. The superiority of algorithms is evaluated by the mean testing accuracies, the stability of algorithms is verified by the standard deviation of testing accuracies which are shown in table 3. From the table, we can see NPHBGrC algorithms are superior to HBGrC algorithms regardless of the maximum testing accuracy (max), the mean testing accuracy (mean), or the minimum testing accuracy (min). On the other hand, we can see from the table that the standard deviations of 10-fold cross validation by NPHBGrC is less than those of HBGrC, which shows that NPHBGrC algorithms are more stable than HBGrC algorithms.

Point 3:  Please check for consistency of formulas: italics, boldface, etc.

Response 3: Thank you very much, we revised the formulas.

Round 2

Reviewer 1 Report

After review of resubmission I recommend this paper be published in its present form.

Reviewer 3 Report

The manuscript has been improved, and I think it could be published on this journal.